# Associations of Total Dietary Quality Score, Dietary Behavior Adherence, and Dietary Portion Adherence with Metabolic Factors Among People with Type 2 Diabetes Mellitus

**DOI:** 10.3390/nu17213366

**Published:** 2025-10-26

**Authors:** Pi-Hui Hsu, Ming-Chieh Tsai, Chiao-Ya Chan, Chih-Yuan Wang, Jung-Fu Chen, Shih-Te Tu, Horng-Yih Ou, Chun-Chuan Lee

**Affiliations:** 1Ancare Nursing Home, Taitung City 950024, Taiwan; 2Department of Medicine, MacKay Medical University, New Taipei City 252005, Taiwan; 3Division of Endocrinology, Department of Internal Medicine, MacKay Memorial Hospital, Taipei City 104217, Taiwan; 4Institute of Epidemiology and Preventive Medicine, National Taiwan University, Taipei City 100225, Taiwan; 5Department of Internal Medicine, National Taiwan University Hospital, College of Medicine, National Taiwan University, Taipei City 100225, Taiwan; 6Division of Endocrinology and Metabolism, Department of Internal Medicine, Kaohsiung Chang Gung Hospital, Chang Gung University College of Medicine, Kaohsiung City 833401, Taiwan; 7Department of Endocrinology and Metabolism, Changhua Christian Hospital, Changhua City 500006, Taiwan; 8Department of Internal Medicine, National Cheng Kung University Hospital, College of Medicine, National Cheng Kung University, Tainan City 704302, Taiwan; mikonlon@gmail.com

**Keywords:** dietary behaviors adherence, dietary portion adherence, total dietary quality scores, type 2 diabetes mellitus

## Abstract

**Aims:** To examine the association between the total dietary quality score (TDQS) and metabolic outcomes among adults with type 2 diabetes (T2DM) in Taiwan. **Methods:** This cross-sectional study enrolled 981 adults with T2DM from 40 diabetes health promotion institutions (DHIPs). Registered dietitians assessed dietary behavior adherence (DBA) and dietary portion adherence (DPA) using a validated dietary quality questionnaire, which were combined into a TDQS. Participants were categorized into tertiles: G1 (≤106.7), G2 (106.8–118.7), and G3 (≥118.8). Associations with metabolic outcomes were analyzed using ANOVA, logistic, and multiple linear regression. **Results**: Participants in the highest TDQS tertile (G3) had significantly lower BMI, waist circumference (WC), fasting blood glucose (FBG), HbA1c, total cholesterol, and triglyceride levels, and a higher proportion achieved HbA1c < 7%. Compared with lower tertiles, G3 participants consumed more vegetables and derived a greater proportion of total energy from protein, whereas participants with a lower TDQS reported higher total energy and fat intakes. Each 1-point increase in TDQS was independently associated with reductions in WC (−0.155 cm), BMI (−0.151 kg/m^2^), FBG (−0.106 mg/dL), HbA1c (−0.136%), total cholesterol (−0.076 mg/dL), and triglyceride levels (−0.148 mg/dL). **Conclusions**: Higher TDQS is significantly associated with improved metabolic outcomes in adults with T2DM, supporting its use as a practical, multidimensional tool for clinical nutrition assessment and personalized dietary intervention.

## 1. Introduction

The global prevalence of diabetes is steadily increasing. According to the International Diabetes Federation (IDF) Diabetes Atlas 11th Edition, approximately 589 million adults aged 20–79 years are projected to be living with diabetes by 2024, accounting for 11.1% of the global population in this age [1]. In Taiwan, an estimated 2.598 million adults are affected [1]. Poor glycemic control is associated with an increased risk of both micro-vascular and macro-vascular complications [2]. To enhance long-term outcomes, Taiwan’s government implemented the DHPI program—a multidisciplinary, team-based care model involving physicians, nurses, and dietitians. Patients receive regular consultations and structured diabetes self-management education from certified diabetes educators (CDEs), particularly nurses and dietitians, to strengthen self-care and reduce complications.

MNT plays a pivotal role in diabetes care [3]. The American Diabetes Association (ADA) recommends that individuals with T2DM receive individualized MNT from a registered dietitian to achieve treatment goals [4]. Despite its importance, adherence to dietary recommendations remains suboptimal, influenced by factors such as limited diabetes-related knowledge, duration since diagnosis of diabetes, comorbidities, gender, age, education level, marital status, and socioeconomic status [5,6,7,8].

Unhealthy dietary patterns—such as diets high in saturated fat (SFA) and low in fiber—are associated with elevated BMI and dyslipidemia [9,10,11]. Frequent consumption of sugary foods and snacks is linked to hyperglycemia, and regular eating out is associated with higher blood glucose, lipid levels, blood pressure, and BMI [12]. To mitigate cardio-metabolic risk, the ADA advises limiting foods rich in SFA (e.g., red meat, full-fat dairy products, butter) and replacing sugar-sweetened beverages (including fruit juices) with water or low/no-calorie alternatives [4]. Nutrition education is a core strategy for improving dietary adherence by enhancing the patients’ knowledge of diabetes-related nutrition [6]. Such interventions have been shown to promote healthier dietary behaviors including increased intake of fruits, vegetables, low-glycemic index (LGI) foods, dietary fiber, and unsaturated fats alongside a reduced consumption of added sugars. These dietary modifications are consistently associated with significant improvements in fasting plasma glucose and insulin sensitivity [12,13]. In a study involving individuals with uncontrolled T2DM, nutrition education was associated with improvements in both dietary behavior and clinical outcomes [14]. Patients who actively participated in nutrition education sessions were significantly more likely to adhere to diabetic meal planning (OR = 2.11) and use the food exchange system (OR = 3.07) [15].

To support diabetes prevention and management, individualized meal planning should take into the account nutrient quality, total caloric intake, and metabolic goals [4]. Food portion size is a key determinant of energy intake, and dietitians provide tailored food portion recommendations; however, adherence to these recommendations—particularly with respect to carbohydrate intake—remains limited [16]. In one study of 411 patients with T2DM, those with poor glycemic control consumed significantly more carbohydrates (251 ± 62 g vs. 213 ± 47 g) and less dietary fiber (16.7 ± 4.5 g vs. 20.5 ± 6.1 g) than those with better glycemic control [17]. Pedersen et al. (2007) demonstrated that the use of a portion control plate in obese patients with T2DM led to greater weight reduction and a significant decrease in diabetes medication dosage after six months [18]. Other research has shown that nutrition education may improve satiety, reduce hunger, and enhance adherence to dietary recommendations [5,19]. Such interventions are associated with increased compliance with healthy dietary patterns, improved adherence to daily dietary recommendations, and reductions in fasting glucose. Moreover, patients with better glycemic control were significantly more likely to meet the recommended dietary intake levels [19,20].

This study aimed to assess the adherence to dietary behaviors and food portion recommendations in individuals with diabetes. It further examined the associations between dietary adherence and metabolic outcomes—including glycemic, blood pressure, blood lipids, and anthropometric measures—and evaluated the potential impact on clinical outcomes.

## 2. Methods

### 2.1. Participants

This cross-sectional study was conducted as part of the fourth nationwide Diabetes Quality Survey in Taiwan. Adults with diabetes (≥18 years) were systematically recruited from 40 DHPIs across urban and rural hospitals and clinics. To avoid enrolling individuals from the same household, every fifth eligible individual with diabetes was invited to participate. If the fifth patient declined, the sixth was approached (only one skip allowed); if both declined, the tenth patient was selected. Eligible participants were T1DM or T2DM who had received care for at least one year and had attended at least one follow-up visit within the preceding 3 to 6 months. Individuals with duplicate enrollment records or those unwilling or unable to comply with the study were excluded.

All participants provided written informed consent after receiving a detailed explanation of the study. A registered dietitian conducted face-to-face interviews to collect data on dietary intake and behaviors using a validated dietary quality questionnaire [21] and reviewed previous dietary recommendations. Demographic data (e.g., age, sex, height, weight) and fasting biochemical data from the most recent three months (e.g., HbA1c, LDL-c and triglyceride levels) were obtained. This study was conducted between 1 January and 31 December 2018, and approved by the Joint Institutional Review Board of the Taiwan Medical Research Ethics Foundation (JIRB No. 17-S-019-1).

### 2.2. Measurement Methods

Prior to the study, all CDE dietitians received a briefing and participated in a methodological discussion. Dietitians conducted face-to-face interviews to complete the validated dietary quality questionnaire [21]. DBA was assessed using a food frequency questionnaire to evaluate the participants’ dietary habits during the preceding week. Items included:(1)Intake of high-sugar foods;(2)LGI carbohydrate (CHO) foods (e.g., dried legumes such as adzuki beans, mung beans, low-fat dairy products, and starchy vegetables such as taro, maize, yam);(3)Consumption of ≥2 servings of whole grains per meal;(4)High-fiber whole grains or tubers (e.g., oats, brown rice, coix seeds, taro, sweet potato, pumpkin);(5)Fish rich in *n*-3 fatty acids (*n*-3 FAs) (e.g., Pacific saury, salmon, mackerel);(6)High-fat cooked foods (e.g., deep-fried foods, cream soups);(7)Foods high in saturated (SFA) or trans fats (trans-FAs) (e.g., high-fat meats, animal skin, whole milk, animal fat, hydrogenated oils, coffee creamer);(8)Frequency of dining out (including three main meals, snacks, and late-night meals);(9)Alcohol consumption exceeding two standard drinks per day for men or one for women;(10)Adherence to a balanced intake across the six major food groups.

Each behavior was rated on a scale from 1 to 5 and standardized to a 10-point score, yielding a total DBA score ranging from 20 to 100, with higher scores indicating better self-management practices.

The DPA was conducted using a 24 h dietary recall to evaluate intake across six major food groups and assess consistency with usual dietary portions. Dietary recommendations from the preceding 3 to 6 months were extracted from medical records. Actual intake was compared with recommended portions, with each food category contributing up to 10 points.

For excessive intake, scores were calculated using the formula:**Score = 10 + ((Recommended − Actual)/Recommended × 10)**.

Scores ≤ 0 were recoded as zero prior to analysis.

For insufficient intake, scores were calculated as:**Score = (Actual/Recommended) × 10**.

For vegetable intake, participants who met or exceeded the recommended amount received the full 10 points, while lower intakes were scored using the standard formula.

The total DPA score ranged from 0 to 60, with higher scores indicating greater adherence to recommended dietary portions.

In accordance with the ADA guidelines, individualized meal planning that accounts for nutrient quality, total caloric intake, and metabolic goals is recommended for the prevention and management of pre-diabetes and diabetes [4]. Accordingly, the TDQS was developed by integrating a dietary behavior frequency questionnaire and a dietary portion questionnaire. The TDQS ranged from 20 to 160 points, with higher scores indicating better overall dietary quality.

### 2.3. Outcome Variables

BMI was defined as weight (kg) divided by height in meters squared (kg/m^2^) and categorized as follows: underweight (<18.5 kg/m^2^), normal weight (18.5 to <24 kg/m^2^), overweight (24 to <27 kg/m^2^), and obese (≥27 kg/m^2^). Abdominal obesity was defined as a WC ≥ 90 cm for men and ≥80 cm for women. Targets for optimal metabolic control (ABC goals) were defined as follows: A, HbA1c < 7%, B, blood pressure < 130/80 mmHg, C, LDL-c < 100 mg/dL. Achievement of ABC targets was defined as simultaneously meeting all three criteria.

### 2.4. Statistical Analysis

Statistical analyses were performed using IBM SPSS Statistics, version 23. The TDQS were categorized into tertiles: Group 1 (G1: ≤106.7 points), Group 2 (G2: 106.8–118.7 points), and Group 3 (G3: ≥118.8 points). Continuous variables meeting the assumption of normality were analyzed using one-way analysis of variance (ANOVA), followed by Bonferroni post hoc comparisons. Gender differences within groups were examined using the independent samples *t*-test. Continuous variables are presented as the mean ± standard deviation (SD). Non-normally distributed continuous variables were analyzed using the Kruskal–Wallis test. Gender differences within groups were assessed using the Mann–Whitney U test. Data are presented as median and interquartile range (IQR). Categorical variables were compared using the χ^2^ test and are presented as frequencies and percentages (n, %). Logistic regression was performed to estimate odds ratios (ORs), 95% confidence intervals (CIs), and Nagelkerke R^2^ values for achieving the recommended targets in body composition, glycemic control, blood pressure, and lipid profiles among participants in Groups 1 and 2, using Group 3 as the reference. To assess multicollinearity, variance inflation factors (VIFs) were evaluated using linear regression models. Multiple linear regression was employed to assess the associations between one-point increases in the TDQS, DBA, and DPA scores and changes in BMI, WC, fasting blood glucose (FBG), HbA1c, total cholesterol, and triglyceride levels. For both logistic and multiple linear regression analyses, two models were constructed. Model 1 was adjusted for sex, age, educational level, smoking status, and physical activity time. Model 2 was further adjusted for diabetes duration, diabetes medication use, and receipt of CDE education, in addition to the variables included in Model 1. A two-sided *p*-value < 0.05 was considered statistically significant.

## 3. Results

A total of 1131 participants were initially enrolled in the study. Of these, 99 participants with missing DBA data and 30 with missing DPA data were excluded. Additionally, 20 participants diagnosed with T1DM and one participant with other unspecified conditions were excluded to ensure consistency in the study population. Ultimately, 981 individuals with T2DM were included in the final analysis. The mean age of participants was 62.0 ± 11.6 years, with an average diabetes duration of 10.9 ± 7.9 years. Males comprised 45.3% of the cohort. A total of 98.4% of individuals were prescribed anti-hyperglycemic agents, with a mean of 2.2 ± 0.9 medications used per patient.

### 3.1. Demographic and Anthropometric Comparison by Group and Sexes

In Table 1, participants were stratified into tertile-based groups according to their TDQS scores: G1 (≤106.7), G2 (106.8–118.7), and G3 (≥118.8). Participants in G1 were significantly younger and exhibited a higher body weight, BMI, WC, and smoking prevalence. They were also less likely to have attained only elementary-level education or below, and reported fewer minutes of physical activity per week compared with those in G2 and G3. The prevalence of obesity and abdominal obesity was lowest among the G3 participants (obesity G1: 50.0%, G2: 40.3%, G3: 29.2%; *p* < 0.001; abdominal obesity G1: 80.2%, G2: 72.3%, G3: 64.6%; *p* < 0.001).

In the sex-stratified analysis within each group, smoking prevalence was consistently higher among males than females. The proportion of males with only elementary education or below was relatively low. Weekly physical activity time was greater among males than females only in G1 (*p* < 0.05), whereas no significant sex differences were observed in G2 and G3. Among the participants in G1, females exhibited a significantly higher BMI compared with males (28.8 ± 5.2 vs. 26.8 ± 4.6 kg/m^2^; *p* < 0.001), along with a greater prevalence of obesity (58.9% vs. 40.4%; *p* < 0.001). Moreover, across all three groups, abdominal obesity was consistently more prevalent among females than males.

### 3.2. Biochemical and Physical Examination Across Groups and Sexes

Participants in G3 also had a longer duration of diabetes (11.8 ± 8.4 years), lower FBG (135 ± 38 mg/dL), and HbA1c (7.7 ± 1.1%). Triglyceride concentrations were also lower, with a median (IQR) of 115 (81) mg/dL. There were no significant differences in the achievement rates of the ABC targets, blood pressure (BP) target, and LDL-c target among the three groups. However, the proportion of participants achieving the HbA1c target was significantly higher in G3 than in G2 and G1 (54.6% vs. 51.3% vs. 43.7%, respectively; *p* = 0.016).

In G3, the DM duration was significantly longer in males than in females (13.8 ± 9.6 vs. 10.4 ± 7.2 years; *p* < 0.005). No significant gender differences were observed across the three groups in terms of BP, FBG, HbA1c, TG, or ABC goal attainment rates. Total cholesterol levels were significantly higher in females than in males in both G3 and G1. HDL-c levels were consistently higher in females across all three groups. LDL-c was significantly higher in females than in males only in G1 (94.4 ± 31.2 vs. 86.6 ± 27.3 mg/dL; *p* < 0.05), resulting in a significantly lower LDL-c target attainment rate among females compared with males (60.7% vs. 74.4%; *p* < 0.05).

### 3.3. Gender Differences in Dietary Behavior Adherence (DBA) Scores

Across all three groups, participants demonstrated relatively low scores in the consumption of “LGI CHO foods”, “high-fiber CHO foods”, and” n-3-rich fish”. In G2, females scored significantly higher than males in the intake of “high-fiber CHO foods” (5.8 ± 2.5 vs. 5.5 ± 2.5; *p* < 0.05). In G1 and G3, females exhibited higher scores than males in the behavior of “less food high in SFAs and TFAs”. Additionally, females in G2 and G3 scored higher than males in the behavior of “dining out less frequently”. Across all three groups, females had significantly higher scores than males in the behavior of “drinking alcohol less frequently”. Conversely, in G3, males scored significantly higher than females in the behavior of maintaining a “balanced diet” (8.8 ± 1.6 vs. 8.3 ± 2.2; *p* < 0.05).

### 3.4. Dietary Intake and Adherence Across Groups and Sexes

Among the six major food categories, the only non-significant difference across the three groups was in the number of servings of protein-rich foods (soy, fish, eggs, and meat). G1 consumed significantly more whole grains (10.4 ± 4.0 servings) and fats/nuts (6.6 ± 3.3 servings), while G3 had the highest vegetable intake (3.6 ± 1.4 servings). Across all groups, male participants had higher intakes of whole grains, soy/fish/egg/meat, and fats/nuts compared with female participants.

Regarding the DPA score, lower compliance was observed in the fruit, dairy, and fat/nuts categories-. In G1, female participants demonstrated significantly higher adherence scores for vegetable and soy/fish/eggs/meat than males. Conversely, in G1 and G2, male participants showed significantly higher adherence scores for dairy products compared with females.

Estimated energy and macronutrient intake were assessed across the three groups. G1 had a significantly higher total energy intake (1694 ± 507 kcal/day) and fat intake (60 ± 24 g/day), while carbohydrate and protein intake did not differ significantly among the groups. In terms of macronutrient distribution as a percentage of total energy intake, only protein showed a significant difference, with G3 exhibiting a higher proportion of energy from protein (14.9 ± 1.6% TE) compared with G1 and G2. Furthermore, G3 had a significantly higher intake per kilogram of current body weight (CBW), with an average of 25.7 ± 5.6 kcal/kg CBW for energy and 1.0 ± 0.2 g/kg CBW for protein, both exceeding the values observed in G1 and G2.

### 3.5. Impact of TDQS on Body Composition and Metabolic Biomarkers

Compared with participants in G3, those in G1 had significantly increased odds of abnormal BW (OR 1.486; 95% CI: 1.032 to 2.139; *p* = 0.033) and abdominal obesity (OR 2.258; 95% CI: 1.528 to 3.336; *p* < 0.001), and achieved HbA1c < 7.0% (OR 0.661; 95% CI: 0.474 to 0.921; *p* = 0.014) and triglyceride levels <150 mg/dL (OR 0.631; 95% CI: 0.443 to 0.899; *p* = 0.011). Participants in G2 also had significantly increased odds of abdominal obesity (OR 1.513; 95% CI: 1.054 to 2.172; *p* = 0.025) (Table 2).

### 3.6. Associations Between TDQS, DBA, and DPA Scores and Metabolic Outcomes

As shown in Table 3, each one-point increase in the TDQS was associated with significant reductions in the WC (−0.155 cm; 95% CI: −0.157 to −0.066; *p* < 0.001), BMI (−0.151 kg/m^2^; 95% CI: −0.064 to −0.026; *p* < 0.001), FBG (−0.106 mg/dL; 95% CI: −0.505 to −0.113; *p* = 0.002), HbA1c (−0.136%; 95% CI: −0.018 to −0.006; *p* < 0.001), total cholesterol (−0.076 mg/dL; 95% CI: −0.327 to −0.017; *p* = 0.029), and triglyceride levels (−0.148 mg/dL; 95% CI: −1.339 to −0.514; *p* < 0.001).

Similarly, each one-point increase in DBA and DPA scores was associated with reductions in WC, BMI, FBG, HbA1c, and triglyceride levels.

## 4. Discussion

### 4.1. TDQS and Metabolic Outcomes

This nationwide, multicenter study involving individuals with T2DM across both urban and rural regions of Taiwan provides robust evidence linking overall dietary quality, measured by the TDQS, to metabolic outcomes. Among the 981 participants (mean age of 62.0 years; mean diabetes duration of 10.9 years), nearly all were receiving anti-hyperglycemic medications, representing a typical middle-aged and older adult T2DM population in Taiwan. Participants who demonstrated greater adherence to dietary behaviors and dietary portion recommendations—reflected by higher TDQS—exhibited significantly more favorable metabolic profiles including lower BMI, WC, FBG, HbA1c, total cholesterol, and triglyceride levels (Table 3) as well as a higher proportion achieving HbA1c < 7%.

The findings of this study demonstrate that adherence to dietary behaviors (DBAs) and portion control (DPA) are significantly associated with metabolic indicators (Table 2 and Table 3), consistent with prior research emphasizing the importance of dietary quality and behavioral modification in the management of T2DM (5,7,20]. For instance, Strydom et al. (2025) showed that personalized nutrition education incorporating glycemic index, glycemic load, and food intake index resulted in significant improvements in glycemic control [20]. Similarly, Tang et al. (2025) and Wilson et al. (2024) highlighted that integrating behavioral modification into dietary interventions enhanced treatment efficacy and facilitated sustained metabolic improvements [5,7]. Collectively, these findings reinforce the notion that comprehensive dietary strategies targeting both behavioral and quantitative aspects of food intake are crucial for optimal diabetes care.

### 4.2. TDQS as a Composite Predictor of Glycemic Control

Diet behaviors and portion-control strategies have consistently been associated with improvements in HbA1c. Schmitt et al. (2013) reported that the Dietary Control subscale of the Diabetes Self-Management Questionnaire (DSMQ) was moderately negatively correlated with HbA1c (r = −0.30) [22]. Similarly, Bukhsh et al. (2017) found that dietary control remained an independent predictor of HbA1c after adjusting for other self-management dimensions (Beta = −0.29, *p* = 0.028) [23]. In the present study, dietary behavior adherence (DBA) predicted HbA1c with Beta = −0.128 (95% CI: −0.028 to −0.009).

Although there is currently no internationally validated dietary portion assessment tool for diabetes, Pedersen et al. (2007) demonstrated that a portion control plate intervention in obese T2DM participants improved HbA1c by 0.5 ± 1.2% [18]. Among patients with type 2 diabetes mellitus (T2DM), a 12-week intervention utilizing a portioned meal box (PMB) significantly alleviated hunger and improved satiety. These effects were associated with reductions in energy, carbohydrate, and fat intake, leading to measurable weight loss and a statistically significant improvement in HbA1c levels [24]. Similarly, Foster et al. (2013) reported a mean HbA1c reduction of 0.7% (95% CI: −0.4 to −1.0%) [25]. In our study, dietary portion adherence (DPA) predicted HbA1c with Beta = −0.087 (95% CI: −0.021 to −0.004). When DBAs and DPA were combined into the TDQS, its predictive ability for HbA1c was Beta = −0.136 (95% CI: −0.018 to −0.006).

### 4.3. Sex Differences in Metabolic Outcomes and Lifestyle

A key finding of this study pertains to sex differences. Female participants exhibited a greater adherence to healthy dietary behaviors—including lower intake of saturated and trans fats, reduced frequency of dining out, lower alcohol consumption, and reduced overall carbohydrate and fat intake—but paradoxically exhibited a higher BMI and a greater prevalence of abdominal obesity than those of the male participants. This discrepancy may be explained by physiological and hormonal factors, particularly postmenopausal declines in estrogen, which promote a redistribution of adipose tissue from the gluteofemoral region to the abdominal area and facilitate visceral fat accumulation [26]. Reduced energy expenditure in postmenopausal women may further exacerbate fat accumulation [27]. Female participants also engaged in fewer minutes of physical activity per week, with those in G1 showing significantly lower activity levels than their male counterparts. According to van Eekelen E (2020), female subjects had significantly higher BMI, hip circumference, total body fat, HDL-c, and inflammatory markers (CRP and TNF). Importantly, increased physical activity was associated with significant reductions in both abdominal and general obesity as well as inflammatory markers among the female participants [28].

Sex differences were also observed in the lipid profiles. Females in G1 exhibited significantly higher LDL-c levels compared with males, despite higher overall HDL-c levels, and demonstrated lower LDL-c target attainment rates. These differences may be attributed to multiple factors, including dietary patterns, medication adherence, and sex-related physiological differences [29,30,31], suggesting potential sex variations in lipid metabolism or treatment response [29,30] that warrant further investigation.

While adherence to healthy dietary patterns generally reduces both the total and visceral fat, the magnitude of improvement appears less pronounced in women than in men [32]. These findings highlight the importance of individualized nutrition counseling for female participants with T2DM, emphasizing energy balance, physical activity, body composition management, and lipid control [27].

### 4.4. Nutritional Intake Patterns, Gut–Brain Axis, and TDQS

Another noteworthy finding is that participants in the high TDQS group (G3) not only consumed greater amounts of vegetables, but also had a higher percentage of energy derived from protein (%TE). Additionally, their protein and caloric intakes per kilogram of body weight were more optimal compared with the other groups. In contrast, participants in G1 exhibited a higher total energy and fat intake. These observations align with prior research demonstrating that increased vegetable consumption is associated with improved metabolic outcomes. For instance, among elderly Japanese patients with T2DM, a daily vegetable intake of ≥150–200 g was associated with lower HbA1c, triglycerides, and WC [33]. Similarly, a 12-week randomized trial in Indonesian adults with T2DM reported that higher raw vegetable intake significantly improved the HbA1c, fasting and postprandial glucose, body weight, WC, and total cholesterol relative to the control group [34]. Conversely, high dietary fat intake has been linked to promoting adiposity, insulin resistance, and unfavorable lipid profiles [35,36]. In our cohort, the elevated fat intake observed in G1 may have contributed to increased adiposity and less favorable glycemic and lipid parameters, underscoring the importance of limiting fat consumption while optimizing protein and vegetable intake.

Emerging evidence further suggests that diet plays a critical role in modulating the gut–brain axis, thereby influencing both metabolic and neurological health [37,38]. Dietary fiber derived from fruits, vegetables, whole grains, and legumes is fermented by gut microbiota to produce short-chain fatty acids (SCFAs). SCFAs stimulate the secretion of glucagon-like peptide-1 (GLP-1) and peptide YY (PYY) from enteroendocrine cells, which collectively suppress appetite and delay gastric emptying. Through gut–brain signaling pathways, SCFAs activate the hypothalamic arcuate nucleus, thereby promoting satiety and regulating host appetite and energy homeostasis [39]. In addition, SCFAs exert anti-inflammatory effects and help maintain intestinal barrier integrity, mechanisms that have been linked to a reduced risk of metabolic syndrome and improved insulin sensitivity [40,41]. Conversely, high-fat diets reduce microbial diversity and promote the proliferation of pathogenic bacteria, contributing to metabolic dysregulation [37,38,42]. Such diets also compromise intestinal barrier integrity, facilitating the translocation of bacterial metabolites—such as lipopolysaccharide (LPS)—into the circulation, thereby triggering chronic low-grade inflammation [43]. Moreover, regular physical activity has been shown to beneficially modulate the gut microbiota by enhancing microbial diversity and stimulating the growth of beneficial bacterial species [37]. Consistent with these mechanisms, participants in G3 who reported longer durations of physical activity and achieved higher DBA scores and DPA scores for vegetables, fruits, and healthy fats (e.g., nuts and seeds) exhibited more favorable metabolic indicators including lower BMI and WC, reduced prevalence of general and abdominal obesity, improved lipid profiles, and better glycemic control. Collectively, these findings support the hypothesis that enhanced dietary quality, as reflected by a higher TDQS, may promote metabolic benefits partly through modulation of the gut–brain axis.

### 4.5. Clinical Implications

McNaughton et al. (2009) demonstrated that dietary quality—measured by adherence to established dietary guidelines—was significantly associated with type 2 diabetes and cardiometabolic risk factors [44]. Building on this evidence, our findings support TDQS as a practical, multidimensional tool for assessing dietary quality and adherence in diabetes care. By integrating behavioral dimensions and food portion components, the TDQS may assist clinicians in identifying patients with suboptimal dietary patterns and guiding individualized nutritional interventions. This approach aligns with the American Diabetes Association’s advocacy of patient-centered and flexible dietary strategies, emphasizing behavioral and individualized nutrition management in diabetes care [45,46].

### 4.6. Limitations

Several limitations of this study should be acknowledged. First, due to its cross-sectional design, causal inferences could not be established. Second, dietary data were primarily obtained through retrospective self-reported questionnaires, which may be subject to recall bias and the underestimation of actual intake. Third, potential interactions involving medication use or energy expenditure were not analyzed, which may influence metabolic outcomes. Fourth, sex-specific interactions with the TDQS were not examined. Fifth, the potential impact of the differing score ranges between DBAs and DPA is acknowledged. As noted, the current TDQS is derived from the direct summation of the two domains without applying weighting adjustments, which may result in a disproportionate contribution from DBAs to the overall score.

Future longitudinal or intervention studies are warranted to verify the causal relationships between the TDQS and metabolic improvements. We recommend that subsequent research examine endpoints such as glycemic control, diabetes-related complications, hospitalization rates, and patient-reported outcomes including quality of life. In addition, future scoring of DBA and DPA should consider standardized or weighted integration methods to enhance interpretability and comparability. The development of a digital health platform based on TDQS may further support routine clinical care by enabling dietary monitoring and personalized nutrition counseling, thereby improving the overall quality of care.

## 5. Conclusions

In conclusion, this nationwide, multicenter study provides robust evidence that a higher TDQS—reflecting adherence to both dietary behaviors and portion recommendations—is significantly associated with improved metabolic outcomes among individuals with T2DM in Taiwan. Participants with a higher TDQS exhibited more favorable profiles in glycemic control, lipid metabolism, and obesity-related parameters. These findings highlight the critical role of comprehensive, patient-centered dietary strategies in diabetes management and support the application of the TDQS as a practical tool for clinical nutrition assessment and personalized intervention planning. Future longitudinal studies are warranted to establish causal relationships and elucidate potential sex-specific responses to dietary interventions.

## Figures and Tables

**Table 1 nutrients-17-03366-t001:** Demographic, anthropometric, biochemical, and dietary quality characteristics of participants with T2DM across TDQS tertiles and by sex within each group.

Characteristics	TDQS G1 (≤106.7)			TDQS G2 (106.8–118.7)			TDQS G3 (≧118.8)			*p* Values Across Groups
	All G1	Male	Female	All G2	Male	Female	All G3	Male	Female	
	N = 324	N = 156	N = 168	N = 318	N = 151	N = 167	N = 339	N = 137	N = 202	
	Mean ± SD	Mean ± SD	Mean ± SD	Mean ± SD	Mean ± SD	Mean ± SD	Mean ± SD	Mean ± SD	Mean ± SD	
Male n (%)	156 (48.1)			151 (47.5)			137 (40.4)			0.09
Age (years)	59.1 ± 12.1 ^a^	59.0 ± 11.1	59.2 ± 12.9	61.7 ± 12.0 ^b^	61.5 ± 12.3	61.9 ± 11.8	65.1 ± 9.8 ^c^	65.4 ± 10.0	64.9 ± 9.7	<0.001
Smoke n (%)			*p* < 0.001			*p* < 0.001			*p* < 0.001	<0.001
None	231 (71.3)	73 (46.8)	158 (94.0)	234 (73.6)	78 (51.7)	156 (93.4)	274 (80.8)	76 (55.5)	198 (98.0)	
Quit smoking	30 (9.3)	29 (18.6)	1 (0.6)	51 (16.0)	43 (28.5)	8 (4.8)	45 (13.3)	41 (29.9)	4 (2.0)	
Smoking	63 (19.4)	54 (34.6)	9 (5.4)	33 (10.4)	30 (19.9)	3 (1.8)	20 (5.9)	20 (14.6)	0 (0.0)	
Education n (%)			*p* < 0.001			*p* < 0.001			*p* < 0.001	0.19
Illiterate	11 (3.4)	0 (0.0)	11 (6.5)	9 (2.8)	2 (1.3)	7 (4.2)	17 (5.0)	2 (1.5)	15 (7.4)	
Elementary	89 (27.5)	34 (21.8)	55 (32.7)	97 (30.5)	32 (21.2)	65 (38.9)	108 (31.9)	31 (22.6)	77 (38.1)	
Junior	71 (21.9)	34 (21.8)	37 (22.0)	41 (12.9)	22 (14.6)	19 (11.4)	59 (17.4)	25 (18.2)	34 (16.8)	
High school	75 (23.2)	43 (27.6)	32 (19.0)	86 (27.0)	41 (27.2)	45 (26.9)	79 (23.3)	31 (22.6)	48 (23.8)	
College	76 (23.5)	43 (27.6)	33 (19.6)	82 (25.8)	54 (35.8)	28 (16.8)	74 (21.8)	47 (43.3)	27 (13.4)	
Unknown	2 (0.6)	2 (1.3)	0 (0.0)	3 (0.9)	0 (0.0)	3 (1.8)	2 (0.6)	1 (0.7)	1 (0.5)	
DM duration (years)	10.1 ± 7.7 ^a^	9.3 ± 6.2	10.8 ± 8.9	10.9 ± 7.5 ^ab^	10.6 ± 7.5	11.1 ± 7.5	11.8 ± 8.4 ^b^	13.8 ± 9.6	10.4 ± 7.2 **	0.02
PAT (mins/week) (median (IQR))	0 (150)	30 (180)	0 (120) *	30 (210)	15 (180)	60 (210)	90 (240)	105 (243)	75 (210)	<0.001
Count of CDE education	6.4 ± 2.7	6.7 ± 2.6	6.1 ± 2.8	6.2 ± 2.8	6.1 ± 2.7	6.1 ± 2.9	6.4 ± 2.7	6.4 ± 2.8	6.5 ± 2.5	0.43
BH (cm)	161 ± 8.8 ^a^	168 ± 6.4	156 ± 6.5 ***	161 ± 8.1 ^a^	167 ± 5.8	156 ± 6.2 ***	159 ± 8.1 ^b^	166 ± 6.7	155 ± 5.2 ***	0.004
CBW (kg)	72.6 ± 14.9 ^a^	75.5 ± 15.3	69.9 ± 14 **	68.8 ± 13.6 ^b^	73.9 ± 13.2	64.2 ± 12.3 ***	64.6 ± 12.3 ^c^	70.7 ± 12.9	60.5 ± 10.1 ***	<0.001
BMI (kg/m^2^)	27.8 ± 5.0 ^a^	26.8 ± 4.6	28.8 ± 5.2 ***	26.4 ± 4.2 ^b^	26.5 ± 4.1	26.3 ± 4.3	25.4 ± 4.0 ^c^	25.5 ± 3.8	25.3 ± 4.1	<0.001
Phenotype n (%)			*p* < 0.001			*p* = 0.60			*p* = 0.76	<0.001
Underweight	0 (0.0)	0 (0.0)	0 (0.0)	6 (1.9)	3 (2.0)	3 (1.8)	8 (2.4)	2 (1.5)	6 (3.0)	
Normal weight	72 (22.2)	45 (28.8)	27 (16.1)	85 (26.7)	35 (23.3)	50 (29.9)	119 (35.1)	46 (33.6)	73 (36.1)	
Overweight	90 (27.8)	48 (30.8)	42 (25.0)	99 (31.1)	50 (33.1)	49 (29.3)	113 (33.3)	48 (35.0)	65 (32.2)	
Obese	162 (50.0)	63 (40.4)	99 (58.9)	128 (40.3)	63 (41.7)	65 (38.9)	99 (29.2)	41 (29.9)	58 (28.7)	
WC (cm)	94.0 ± 11.7 ^a^	94.7 ± 11.5	93.2 ± 12	91.0 ± 10.5 ^b^	93.3 ± 10.2	89.0 ± 10.6 ***	87.7 ± 9.3 ^c^	90.7 ± 9.1	85.6 ± 8.9 ***	<0.001
Abdominal obesity n (%)	260 (80.2)	108 (69.2)	152 (90.5) ***	230 (72.3)	95 (62.9)	135 (80.8) **	87 (64.6)	74 (54.0)	145 (71.8) **	<0.001
SBP (mmHg)	132 ± 18	130 ± 16	133 ± 20	133 ± 16	133 ± 16	134 ± 16	133 ± 17	133 ± 17	133 ± 17	0.54
DBP (mmHg)	76 ± 13	76 ± 11	75 ± 15	76 ± 11	76 ± 12	76 ± 11	75 ± 11	75 ± 9	74 ± 12	0.35
FBG (mg/dL)	148 ± 54 ^a^	143 ± 46	152 ± 60	135 ± 37 ^bc^	133 ± 34	137 ± 39	135 ± 38 ^c^	134 ± 35	136 ± 40	<0.001
HbA1c (%)	7.6 ± 1.6 ^a^	7.5 ± 1.5	7.7 ± 1.7	7.3 ± 1.3 ^bc^	7.3 ± 1.4	7.3 ± 1.2	7.1 ± 1.1^b c^	7.2 ± 1.1	7.1 ± 1.1	<0.001
Total cholesterol (mg/dL)	163 ± 38	158 ± 35	168 ± 40 *	157 ± 34	154 ± 35	159 ± 34	158 ± 35	149 ± 30	164 ± 32 ***	0.05
Triglyceride (mg/dL) (median (IQR))	130 (103)	123 (104)	132 (106)	125 (70)	118 (75)	131 (81)	115 (81)	113 (82)	120 (80)	<0.001
HDL-c (mg/dL) (median (IQR))	45.0 (17.4)	42.0 (16.0)	46.0 (19.0) *	44.0 (16.0)	42.0 (15.0)	46 (18.6) **	46.0 (16.0)	41.2 (13.0)	49.0 (17.0) ***	0.108
LDL-c (mg/dL)	90.6 ± 29.6	86.6 ± 27.3	94.4 ± 31.2 *	89.0 ± 27.5	89.4 ± 28.6	88.7 ± 26.6	87.2 ± 24.1	85.0 ± 22.8	88.8 ± 24.9	0.30
Achieving ABC targets n (%)	46 (14.2)	26 (16.7)	20 (11.9)	47 (14.8)	25 (16.6)	22 (13.2)	52 (15.3)	15 (10.9)	37 (18.3)	0.79
A: HbA1c < 7% n (%)	141 (43.7)	73 (47.1)	68 (40.5)	163 (51.3)	75 (49.7)	88 (52.7)	185 (54.6)	67 (48.9)	118 (58.4)	0.016
B: BP <130/80 mmHg n (%)	128 (39.5)	65 (41.7)	63 (37.5)	115 (36.2)	54 (35.8)	61 (36.5)	122 (36.0)	43 (31.4)	79 (39.1)	0.54
C: LDL-c <100 mg/dL n (%)	218 (67.3)	116 (74.4)	102 (60.7) *	214 (67.3)	102 (67.5)	112 (67.1)	244 (72.0)	104 (75.9)	140 (69.3)	0.26
**DBA score**	61.7 ± 8.0 ^a^	60.9 ± 8.2	62.3 ± 7.7	69.7 ± 6.1 ^b^	68.8 ± 6.2	70.6 ± 5.9 *	77.3 ± 6.3 ^c^	76.8 ± 6.3	77.6 ± 6.4	<0.001
**Items of DBA**										
Less high sugar food	7.1 ± 2.3 ^a^	7.3 ± 2.1	7.0 ± 2.4	7.8 ± 2.0 ^bc^	7.9 ± 1.9	7.7 ± 2.1	8.1 ± 1.8 ^c^	7.9 ± 1.8	8.3 ± 1.8	<0.001
LGI CHO food	4.3 ± 1.9 ^a^	4.2 ± 1.9	4.3 ± 2.0	4.9 ± 2.0 ^b^	4.8 ± 1.9	5.0 ± 2.1	5.7 ± 2.3 ^c^	5.7 ± 2.4	5.7 ± 2.3	<0.001
CHO spacing	8.0 ± 2.5 ^a^	8.0 ± 2.6	8.1 ± 2.5	8.6 ± 1.9 ^b^	8.8 ± 1.9	8.4 ± 1.9	9.1 ± 1.4 ^c^	9.2 ± 1.5	9.0 ± 1.4	<0.001
High fiber CHO food	4.3 ± 2.1 ^a^	4.2 ± 2.1	4.4 ± 2.1	5.5 ± 2.5 ^b^	5.2 ± 2.5	5.8 ± 2.5 *	6.6 ± 2.6 ^c^	6.8 ± 2.7	6.5 ± 2.5	<0.001
Rich n-3 FAs fish	4.5 ± 2.2 ^a^	4.7 ± 2.1	4.4 ± 2.2	5.0 ± 2.3 ^b^	5.1 ± 2.2	4.9 ± 2.3	5.8 ± 2.6 ^c^	6.0 ± 2.5	5.7 ± 2.6	<0.001
Less high fat cooking	6.8 ± 2.4 ^a^	6.8 ± 2.4	6.7 ± 2.4	7.6 ± 2.1 ^b^	7.5 ± 2.2	7.7 ± 2.0	8.2 ± 1.8 ^c^	8.2 ± 1.7	8.1 ± 1.9	<0.001
Less high SFA or TFA food	6.6 ± 2.3 ^a^	6.2 ± 2.2	6.9 ± 2.3 *	7.2 ± 2.0 ^b^	7.0 ± 2.0	7.3 ± 1.9	7.9 ± 2.0 ^c^	7.5 ± 2.2	8.1 ± 1.9 *	<0.001
Less dining out frequency	5.4 ± 2.8 ^a^	5.2 ± 2.8	5.6 ± 2.8	6.3 ± 2.8 ^b^	5.7 ± 2.9	6.9 ± 2.6 ***	7.7 ± 2.4 ^c^	7.3 ± 2.6	8.0 ± 2.1 *	<0.001
Less alcohol drinking	9.0 ± 2.2 ^a^	8.5 ± 2.5	9.5 ± 1.6 ***	9.4 ± 1.5 ^bc^	9.2 ± 1.8	9.7 ± 1.1 **	9.7 ± 1.0 ^c^	9.4 ± 1.5	9.9 ± 0.5 **	<0.001
Balanced diet	5.6 ± 2.7 ^a^	5.7 ± 2.7	5.5 ± 2.7	7.4 ± 2.3 ^b^	7.6 ± 2.3	7.3 ± 2.3	8.5 ± 2.0 ^c^	8.8 ± 1.6	8.3 ± 2.2 *	<0.001
**DPA score**	33.8 ± 7.2 ^a^	33.4 ± 7.8	34.3 ± 6.7	42.9 ± 6.4 ^b^	43.8 ± 6.5	42.0 ± 6.2 *	50.8 ± 6.3 ^c^	51 ± 5.8	50.6 ± 6.7	<0.001
**Six major food groups intake**										
Whole grains and tubers (S)	10.4 ± 4.0 ^a^	11.3 ± 4.3	9.6 ± 3.5 ***	9.5 ± 3.0 ^b^	10.7 ± 3.1	8.4 ± 2.4 ***	9.1 ± 2.4 ^b^	10 ± 2.5	8.5 ± 2.2 ***	<0.001
Fruits (S) (median (IQR))	1.3 (1.7)	1.5 (2.0)	1.0 (1.0)	2.0 (1.5)	2.0 (1.0)	2.0 (2.0)	2.0 (1.0)	2.0 (1.0)	2.0 (0.5)	<0.001
Vegetables (S)	2.4 ± 1.4 ^a^	2.3 ± 1.4	2.4 ± 1.4	3.0 ± 1.5 ^b^	3.0 ± 1.4	3.1 ± 1.5	3.6 ± 1.4 ^c^	3.6 ± 1.5	3.6 ± 1.4	0.02
Dairy products (S) (median (IQR))	0.0 (0.0)	0.0 (0.0)	0.0 (0.0)	0.0 (1.0)	0.0 (0.5)	0.0 (1.0)	0.5 (1.0)	0.8 (1.0)	0.5 (1.0)	<0.001
Soy, fish, eggs, and meat (S)	5.2 ± 2.5	5.9 ± 2.8	4.5 ± 1.9 ***	4.9 ± 1.9	5.4 ± 1.9	4.5 ± 1.8 ***	4.9 ± 1.5	5.2 ± 1.6	4.7 ± 1.5 ***	0.12
Oils and nuts (S)	6.6 ± 3.3 ^a^	7.0 ± 3.5	6.2 ± 3.0 *	6.1 ± 2.6 ^b^	6.5 ± 2.6	5.7 ± 2.5 *	5.9 ± 2.1 ^b^	6.3 ± 2.4	5.6 ± 1.9 *	0.002
**Six major food groups score**										
Whole grains and tubers	7.5 ± 2.1 ^a^	7.4 ± 2.1	7.6 ± 2.0	8.1 ± 1.8 ^b^	8.1 ± 1.8	8.0 ± 1.8	8.8 ± 1.5 ^c^	8.8 ± 1.3	8.8 ± 1.6	<0.001
Fruits	4.3 ± 3.7 ^a^	4.3 ± 3.7	4.4 ± 3.7	5.9 ± 3.8 ^b^	5.8 ± 3.7	5.9 ± 4.0	7.9 ± 2.9 ^c^	7.7 ± 2.9	8.0 ± 2.8	<0.001
Vegetables	6.0 ± 2.9 ^a^	5.6 ± 2.7	6.3 ± 3.0 *	7.6 ± 2.4 ^b^	7.5 ± 2.4	7.6 ± 2.4	8.8 ± 1.9 ^c^	8.8 ± 1.9	8.7 ± 1.9	<0.001
Dairy products	4.2 ± 4.8 ^a^	4.8 ± 4.9	3.7 ± 4.6 *	6.8 ± 4.4 ^b^	7.4 ± 4.2	6.3 ± 4.5 *	8.5 ± 3.3 ^c^	8.6 ± 3.3	8.4 ± 3.4	<0.001
Soy, fish, egg and meat	6.6 ± 2.6 ^a^	6.2 ± 2.8	6.9 ± 2.3 *	7.5 ± 2.3 ^b^	7.7 ± 2.3	7.3 ± 2.4	8.4 ± 1.9 ^c^	8.6 ± 1.6	8.3 ± 2.1	<0.001
Oils and nuts	5.2 ± 3.2 ^a^	5.0 ± 3.2	5.3 ± 3.2	7.1 ± 2.9 ^b^	7.3 ± 2.7	6.9 ± 3.0	8.4 ± 2.0 ^c^	8.5 ± 1.9	8.4 ± 2.1	<0.001
**Total energy intake** (kcal/day)	1694 ± 507	1846 ± 530	1554 ± 443 ***	1630 ± 373	1759 ± 347	1512 ± 357 ***	1622 ± 285	1739 ± 282	1542 ± 259 ***	0.04
CHO (g/day)	197 ± 68	211 ± 71	184 ± 62 ***	192 ± 51	208 ± 50	178 ± 48 ***	189 ± 39	202 ± 42	179 ± 35 ***	0.15
CHO (% TE)	46.8 ± 9.2	46.1 ± 9.6	47.5 ± 8.8	47.3 ± 7.4	47.3 ± 7.3	47.3 ± 7.6	46.7 ± 6.2	46.6 ± 6.8	46.7 ± 5.9	0.51
Protein (g/day)	61 ± 21	68 ± 22	55 ± 18 ***	59 ± 16	64 ± 15	54 ± 15 ***	60 ± 13	65 ± 12	57 ± 12 ***	0.24
Protein (% TE)	14.4 ± 2.3 ^a^	14.8 ± 2.3	14.1 ± 2.3 *	14.5 ± 1.8 ^a^	14.6 ± 1.8	14.4 ± 1.8	14.9 ± 1.6 ^b^	15 ± 1.4	14.8 ± 1.7	0.008
Fat (g/day)	60 ± 24 ^a^	66 ± 26	54 ± 21 ***	56 ± 18 ^b^	60 ± 18	52 ± 17 ***	56 ± 14 ^b^	60 ± 16	53 ± 13 ***	0.02
Fat (% TE)	31.5 ± 7.9	31.7 ± 8.1	31.3 ± 7.7	30.9 ± 6.6	30.7 ± 6.3	31.1 ± 6.8	31 ± 5.4	31 ± 6.2	31.1 ± 4.8	0.94
Energy intake (kcal/kg CBW)	23.9 ± 7.6 ^a^	25.1 ± 8.1	22.8 ± 7.0 *	24.3 ± 6.2 ^a^	24.4 ± 5.8	24.2 ± 6.5	25.7 ± 5.6 ^b^	25.2 ± 5.3	26.1 ± 5.8	0.001
Protein intake (g/kg CBW)	0.9 ± 0.3 ^a^	0.9 ± 0.3	0.8 ± 0.3 ***	0.9 ± 0.2 ^a^	0.9 ± 0.2	0.9 ± 0.3	1.0 ± 0.2 ^b^	0.9 ± 0.2	1.0 ± 0.3	<0.001

Continuous variables with normal distribution are presented as the mean ± SD, while non-normally distributed continuous variables are presented as the median (IRQ). Categorical variables are reported as n (%). Group comparisons across TDQS tertiles were performed using one-way ANOVA with Bonferroni post hoc tests for continuous variables and chi-square tests for categorical variables. The Kruskal–Wallis test was used to analyze non-normally distributed continuous variables, while gender differences within groups were assessed using the Mann–Whitney U test due to non-normal distribution. Sex-specific comparisons within each TDQS group were assessed using independent-samples *t*-tests for continuous variables and chi-square tests for categorical variables. Superscript letters (a, b, c) indicate significant differences among TDQS groups (*p* < 0.05). Asterisks indicate sex-specific statistical significance: * *p* < 0.05; ** *p* < 0.005; *** *p* < 0.001. Abbreviations: BH, body height; CBW, current body weight; PAT, physical activity time; WC, waist circumference; BMI, body mass index; SBP: systolic blood pressure; DBP: diastolic blood pressure; FBG, fasting blood glucose; HDL-c, high-density lipoprotein cholesterol; LDL-c, low-density lipoprotein cholesterol; BP, blood pressure. S, serving; TE, total energy; IBW, ideal body weight.

**Table 2 nutrients-17-03366-t002:** Effect of the TDQS on body weight, waist circumference, HbA1c, and triglyceride levels (data referenced from G3).

Outcome		TDQS G1 (≤106.7)					TDQS G2 (106.8–118.7)					Nagelkerke R^2^	VIF
		OR	95% CI			*p*-Value	OR	95% CI			*p*-Value		
Abnormal BW	Model 1	1.530	1.067	-	2.192	0.021	1.333	0.946	-	1.878	0.100	0.061	1.080
	Model 2	1.486	1.032	-	2.139	0.033	1.297	0.917	-	1.835	0.141	0.064	1.084
Abdominal obesity	Model 1	2.282	1.553	-	3.355	0.000	1.545	1.081	-	2.208	0.017	0.116	1.079
	Model 2	2.258	1.528	-	3.336	0.000	1.513	1.054	-	2.172	0.025	0.118	1.084
HbA1C	Model 1	0.706	0.513	-	0.971	0.032	0.917	0.672	-	1.253	0.588	0.019	1.079
	Model 2	0.661	0.474	-	0.921	0.014	0.906	0.656	-	1.252	0.550	0.080	1.084
triglyceride	Model 1	0.619	0.438	-	0.876	0.007	0.794	0.561	-	1.124	0.193	0.050	1.082
	Model 2	0.631	0.443	-	0.899	0.011	0.831	0.583	-	1.183	0.304	0.053	1.086

The effect of the TDQS was assessed using logistic regression analysis, with outcome variables dichotomized based on clinically relevant thresholds. Variance inflation factors (VIFs) were evaluated using linear regression to examine multicollinearity. Model 1: adjusted for sex, age, educational level, smoking status, and physical activity. Model 2: adjusted for age, sex, education level, smoking status, physical activity, diabetes duration, diabetes medication use, and receipt of CDE education.

**Table 3 nutrients-17-03366-t003:** Prediction of metabolic outcomes by the TDQS, DBA score, and DPA score.

Exposure	Metabolic Outcome	Model 1							Model 2						
		Beta	95% CI			*p*-Value	R^2^	VIF	Beta	95% CI			*p*-Value	R^2^	VIF
TDQS	WC (cm)	−0.159	−0.158	-	−0.069	0.000	0.111	1.105	−0.155	−0.157	-	−0.066	0.000	0.115	1.109
	BMI (kg/m^2^)	−0.153	−0.064	-	−0.027	0.000	0.111	1.106	−0.151	−0.064	-	−0.026	0.000	0.117	1.109
	FBG (mg/dL)	−0.104	−0.493	-	−0.106	0.002	0.024	1.107	−0.106	−0.505	-	−0.113	0.002	0.036	1.111
	HbA1c (%)	−0.131	−0.018	-	−0.006	0.000	0.040	1.105	−0.136	−0.018	-	−0.006	0.000	0.079	1.109
	TC (mg/dL)	−0.079	−0.332	-	−0.027	0.021	0.031	1.109	−0.076	−0.327	-	−0.017	0.029	0.030	1.112
	TG (mg/dL)	−0.152	−1.351	-	−0.540	0.000	0.062	1.109	−0.148	−1.339	-	−0.514	0.000	0.065	1.113
DBA score	WC (cm)	−0.107	−0.199	-	−0.049	0.001	0.098	1.157	−0.102	−0.194	-	−0.043	0.002	0.102	1.160
	BMI (kg/m^2^)	−0.099	−0.079	-	−0.017	0.003	0.098	1.160	−0.097	−0.078	-	−0.015	0.004	0.104	1.163
	FBG (mg/dL)	−0.091	−0.746	-	−0.106	0.009	0.021	1.162	−0.086	−0.722	-	−0.078	0.015	0.033	1.162
	HbA1c (%)	−0.135	−0.029	-	−0.010	0.000	0.040	1.159	−0.128	−0.028	-	−0.009	0.000	0.077	1.162
	TC (mg/dL)	−0.087	−0.578	-	−0.068	0.013	0.032	1.154	−0.087	−0.581	-	−0.064	0.014	0.032	1.160
	TG (mg/dL)	−0.141	−2.103	-	−0.753	0.000	0.059	1.161	−0.137	−2.068	-	−0.700	0.000	0.061	1.164
DPA score	WC (cm)	−0.140	−0.228	-	−0.089	0.000	0.107	1.020	−0.138	−0.227	-	−0.087	0.000	0.112	1.024
	BMI (kg/m^2^)	−0.139	−0.094	-	−0.037	0.000	0.108	1.020	−0.136	−0.094	-	−0.036	0.000	0.114	1.024
	FBG (mg/dL)	−0.073	−0.627	-	−0.038	0.027	0.019	1.019	−0.081	−0.671	-	−0.074	0.014	0.033	1.023
	HbA1c (%)	−0.075	−0.020	-	−0.002	0.019	0.030	1.020	−0.087	−0.021	-	−0.004	0.006	0.070	1.024
	TC (mg/dL)	−0.041	−0.379	-	0.083	0.210	0.027	1.024	−0.036	−0.364	-	0.105	0.279	0.026	1.027
	TG (mg/dL)	−0.100	−1.599	-	−0.364	0.002	0.051	1.022	−0.098	−1.593	-	−0.336	0.003	0.055	1.026

Metabolic outcomes were predicted using multiple linear regression, with outcome variables treated as continuous measures. Model 1: adjusted for sex, age, educational level, smoking status, and physical activity. Model 2: adjusted for age, sex, education level, smoking status, physical activity, diabetes duration, diabetes medication use, and receipt of CDE education. Abbreviations: WC, waist circumference; BMI, body mass index; FBG, fasting blood glucose; TC, total cholesterol; TG, triglyceride.

## Data Availability

Restrictions apply to the availability of some or all data generated or analyzed during this study to preserve patient confidentiality or because they were used under license. The corresponding author will, on request, detail the restrictions and any conditions under which access to some data may be provided.

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
