# Peer review of "Associations of Total Dietary Quality Score, Dietary Behavior Adherence, and Dietary Portion Adherence with Metabolic Factors Among People with Type 2 Diabetes Mellitus"

_nutrients, 2025, doi:10.3390/nu17213366_

Round 1

Reviewer 1 Report

Comments and Suggestions for Authors

General Comments

It is unclear what the purpose of the summary scores are. To obtain the scores it is necessary for the clinician/health worker to measure all of the component dietary categories. So there is no cost/time saving in using the DBA, DPA, TDQS scores as opposed to examining the items on which they are based. However, using these scores incurs a cost in lost information. If DBA is low, indicating a poor diet, then what is the indication to ameliorate the poor diet? The clinician would have to look at the components anyway to see what has to be changed to improve the diet. There is also the possibility that a poor diet will give an average DPA, that is, a DPA score that indicates no dieting problems. For example, suppose 3 food categories have portions below the recommended amounts while 3 categories have portions in excess of recommendations. According to the DPA scoring algorithm this will likely indicate no net gain or loss in DPA, hence no dietary problems. Lastly, why have a TDQS (=DBA + DPA)? What does TDQS add beyond DBA and DPA? The authors need to provide a much stronger motivation for these summary diet scores.

Are DBA and DPA preexisting scores?  If so then the authors need to provide references for them including references on validation of these scores.  If they are not preexisting scores then the authors need to provide more detail on calculating DBA. “Each behavior was scored individually” (#118) provides no information on how the scoring was done. Since there are 10 DBA items listed in Table 1, we may assume that each item is scored 0 – 10 to achieve the maximum of 100 but there is no indication of how a score is assigned. Does a “10” for “Less high sugar food” mean no high sugar food or just at or below recommended amounts? From the provided description it is impossible for the reader to calculate DBA scores for their own data and therefore provide a replication this study. Also, if the DBA and DPA scores are not preexisting then there needs to be some validation of the measures. As mentioned in the prior section on motivation, it is possible for the DPA to fail to indicate a dietary problem of over and under recommendation scores leading to a zero net gain/loss. This suggests that DPA may not be a good indicator of adherence to recommended dietary proportions. The authors need to provide references or far more detail for their main measures.

There are 20 T1DM participant and 1 Other (which is not described anywhere … MODY?). Given that these have a different clinical trajectory (e.g., onset is much younger and treatment includes insulin much sooner) compared to T2DM and there are so very few then it would make for a much cleaner study to exclude these.

[Methods]

Did any of the participants have a shared household? For example, was the wife of a male participant also included in the study? If a husband and wife pair were included in the study then their dietary measures would be dependent to some extent. This would violate the assumptions of independent measures used in most of the statistical analyses used.

[Outcome variables]

This paragraph is confusing. 

(1) How is BMI in 4 categories an outcome variable for either logistic regression (requires a dichotomous outcome) or multiple linear regression (requires a continuous outcome).

(2) under Outcome Variables, WC, HbA1c, LDL-c, etc. are listed as above/below threshold dichotomous outcomes but under Statistical Analysis they are used in Pearson correlation analyses that expect continuous measures. Obviously, the authors used the continuous measures without thresholds but that is not what the Outcome Variables paragraph indicates. The authors need to carefully revise the Outcome Variables paragraph to more accurately state the outcome variables they used. For example, a description along the lines of “In addition to being used as continuous variables, metabolic measures were also used as dichotomous outcomes based on the thresholds: HbA1c <7%, … “ etc.

[Statistical Analyses]

There is a very large number of variables listed in Table 1 resulting in a large number of male vs female hypotheses tested. This is then repeated for TDQS tertiles G1/G2/G3. And there are even more hypotheses considered. The authors should consider some adjustment for the multiple hypothesis testing; at the very least the authors need to discuss the false positive problem.

[Results]

There is a confound in these analyses between sex and TDQS tertile. For example, 95.4% of females are “None” for smoking compared to 51.3% of males (a significant difference). TDQS G3 has more females (60.1%) compared to G2 (51.9%) or G1 (52.1%) so it is not unexpected that G3 has significantly more Smoking “None” (82.2%) than G2 (73.0%) or G1 (71.3%). The problem is that we cannot tell whether the significant difference in smoking between TDQS tertiles is due to the adherence (or not) to dietary guidelines or is due to a significant difference between males and females and the TDQS tertiles do not have equal proportions of each. The example of Smoking “None” was only chosen because it makes the issue clear but this is a factor for all measures that are significantly different between males and females, and that includes DBA.  Did the authors look at separate male and female analyses, that is, separate Table 3s for males and females? If so then was there any indication of different results for males compared to females indicative of a ex by TDQS interaction?

[Table 1]

The authors need to define “Achieving ABC targets”.  Does this mean that all three ABD targets were achieved, or just that at least one ABC target was achieved, or what? The authors need to clarify this somewhere.

The authors need to define “CHO” somewhere. Is there a difference between “CHO food” and “Carb food”?

[Table 3]

“a, b Different marker indicate significant differences among the groups.” This is not very informative. The authors need to be more specific.

Author Response

Dear Reviewer,

Thank you very much for your valuable suggestions and guidance. 

Please find our point-by-point responses in the attached document.

We sincerely appreciate your time and thoughtful consideration.

With sincere appreciation.

Pi-Hui Hsu

2025/10/18

Reviewer 2 Report

Comments and Suggestions for Authors

The manuscript presents a nationwide cross-sectional study examining the association between dietary behavior adherence (DBA), dietary portion adherence (DPA), and metabolic outcomes in adults with diabetes in Taiwan. The study addresses a clinically relevant topic and utilizes a large, representative sample. However, several methodological, statistical, and structural issues must be addressed to improve the manuscript’s clarity, validity, and scientific rigor.

  1. Scientific Rationale and Novelty
  • The integration of DBA and DPA into a composite Total Dietary Quality Score (TDQS) is conceptually interesting, but the rationale for combining these domains is not clearly justified. Are they equally weighted in clinical relevance?
  • The DBA and DPA scoring systems appear to be locally developed and lack reference to validation studies or international comparability. Please clarify their origin and whether they have been validated in previous research.
  • While similar studies have explored dietary adherence and metabolic outcomes, the use of TDQS as a composite index in a diabetes-specific population may offer novel insights. However, the manuscript should better position its contribution relative to existing literature.
  1. Study Design and Confounding
  • The cross-sectional design inherently limits causal inference. This limitation should be explicitly acknowledged in the Discussion.
  • Key confounders such as age, gender, medication use, physical activity, smoking, alcohol consumption, and socioeconomic status were either not adjusted for or not clearly reported in the regression models. Their omission raises concerns about the internal validity of the observed associations.
  • Although model 2 includes some demographic variables, it remains unclear whether medication use and other lifestyle factors were considered. Please clarify and, if possible, reanalyze with appropriate adjustments.
  • The classification of diabetes type (T1DM vs. T2DM) is not well defined. Given the distinct pathophysiology and dietary management of T1DM and T2DM, consider conducting a sensitivity analysis excluding T1DM participants.
  1. Measurement and Statistical Transparency
  • It is unclear whether metabolic parameters (e.g., HbA1c, triglycerides) were measured under fasting conditions. This is critical for interpreting the results.
  • The use of Ideal Body Weight (IBW) is mentioned, but no reference or justification is provided for its application over other metrics such as actual body weight or BMI.
  • Tables present data as mean ± SD, but this format is not consistently indicated. Please specify whether variables were normally distributed and consider using median/IQR for skewed data.
  • Confidence intervals (CI) are only reported in logistic regression tables. Including CI in linear regression outputs would enhance interpretability.
  • Model fit statistics (e.g., R², AUC) and multicollinearity diagnostics (e.g., VIF) are missing. Including these would strengthen the statistical rigor.
  1. Results Presentation
  • The Results section is disproportionately brief compared to the extensive tables (5 tables spanning 13 pages). Many findings are buried in the tables and not adequately discussed in the text.
  • Consider condensing the tables, merging overlapping content, and moving supplementary details to an appendix. Expand the narrative description to highlight key findings and their implications.
  1. Interpretation and Clinical Relevance
  • While statistically significant associations are reported (e.g., −0.012% HbA1c per TDQS point), their clinical relevance is unclear. Are these changes meaningful in terms of patient outcomes or treatment decisions?
  • The paradoxical finding that women had higher TDQS but also higher abdominal obesity warrants deeper exploration. Potential hormonal, behavioral, or cultural explanations should be discussed.
  • The manuscript should better contextualize its findings within international literature and discuss generalizability beyond Taiwan.
  1. Limitations
  • The manuscript lacks a dedicated Limitations section. Key issues such as cross-sectional design, self-reported dietary data, lack of adjustment for confounders, and potential selection bias (e.g., recruitment from DHPI centers) should be explicitly acknowledged.
  • The absence of validated dietary adherence tools and the reliance on institution-specific scoring systems should also be discussed as a limitation.
  1. Future Directions
  • The authors mention the need for prospective studies. It would be helpful to outline specific hypotheses or endpoints for future research, such as diabetes complications, hospitalization rates, or quality of life.
  • Consider discussing how TDQS could be integrated into routine clinical practice or digital health platforms for dietary monitoring.

Overall Recommendation

This manuscript has potential to contribute meaningfully to the literature on dietary adherence and metabolic health in diabetes care. However, major revisions are needed to clarify methodology, strengthen statistical analysis, and improve the structure and interpretability of the results.

Author Response

(The authors gave the same response as above.)

Reviewer 3 Report

Comments and Suggestions for Authors

In the This cross-sectional study, the authors found that t higher TDQS are significantly as
sociated with lower BMI, WC, HbA1c, and triglyceride levels, highlighting the pivotal role
of comprehensive dietary adherence in diabetes management. The aim of the study and initial hypothesis are clear. The manuscript is highly relevant and adequately presents the literature, including recognition of gaps in knowledge, interpretation of findings, and the significance of other recent research on the topic. lthough the findings are impressive, I would like to make the folllowing comments:

  1. It remaines unclear how the authors link TDQS with portion adequacy, energy value and adherence to eating.
  2. The associations between higher TDQS and improved anthropometric and biochemical biomarkers require explanation in connection with gut-brain metabolic axis. Please, hypothesize and extensively report approapriate issues.
  3. Conclusion: nutrition education is unlikely to be clear term in this context. Please, provide the concise conclusion in thorough relation with the data received.

Author Response

(The authors gave the same response as above.)

Round 2

Reviewer 2 Report

Comments and Suggestions for Authors

Thank you for the detailed and thoughtful revisions. I have reviewed the authors’ responses and the updated manuscript, and I find that the concerns raised have been adequately addressed.